# Reversal of transmission and reflection based on acoustic metagratings with integer parity design

Yangyang Fu[1,2,6], Chen Shen [iD] [3,6], Yanyan Cao[1,6], Lei Gao[1], Huanyang Chen[4], C.T. Chan[5], Steven A. Cummer[3] & Yadong Xu[1]

Phase gradient metagratings (PGMs) have provided unprecedented opportunities for wavefront manipulation. However, this approach suffers from fundamental limits on conversion efficiency; in some cases, higher order diffraction caused by the periodicity can be observed distinctly, while the working mechanism still is not fully understood, especially in refractive-type metagratings. Here we show, analytically and experimentally, a refractive-type metagrating which can enable anomalous reflection and refraction with almost unity efficiency over a wide incident range. A simple physical picture is presented to reveal the underlying diffraction mechanism. Interestingly, it is found that the anomalous transmission and reflection through higher order diffraction can be completely reversed by changing the integer parity of the PGM design, and such phenomenon is very robust. Two refractive acoustic metagratings are designed and fabricated based on this principle and the experimental results verify the theory.

[1] School of Physical Science and Technology and Jiangsu Key Laboratory of Thin Films, Soochow University, Suzhou 215006, China. [2] College of Science, Nanjing University of Aeronautics and Astronautics, Nanjing 211106, China. [3] Department of Electrical and Computer Engineering, Duke University, Durham, North Carolina 27708, USA. [4] Institute of Electromagnetics and Acoustics and Key Laboratory of Electromagnetic Wave Science and Detection Technology, Xiamen University, Xiamen 361005, China. [5] Department of Physics, Hong Kong University of Science and Technology, Clear Water Bay, Hong Kong, China. [6]These authors contributed equally: Yangyang Fu, Chen Shen, Yanyan Cao. Correspondence and requests for materials should be addressed to C.T.C. (email: phchan@ust.hk) or to S.A.C. (email: cummer@duke.edu) or to Y.X. (email: ydxu@suda.edu.cn)

The ability to control at will the propagation of waves, such as electromagnetic waves and acoustic waves, has captured the fascination of scientists. In the past few years, as the 2D version of bulk metamaterials, metasurfaces have provided new paradigms to build devices that direct the flow of waves in a way not possible before[1–4], and have enabled new physics[5–9] that are distinctly different from those observed in their 3D counterparts (i.e., metamaterials). Typical examples, include planar lenses[5], holograms[6], ultrathin cloaking[7] in electromagnetics, and other devices in acoustics[10,11]. By engineering phase shift $\phi(x)$ along metasurfaces, the scattered wavefronts can be manipulated to achieve anomalous reflection or refraction[12–16], which is summarized as the generalized Snell's law (GSL)[12],

$$k_x^{in} = k_x^{r(t)} - \xi, \qquad (1)$$

where $k_x^{in}$ and $k_x^{r(t)}$ are tangential wave vectors of incident and reflected (transmitted) wave. For the 2D case, $\xi = \partial\phi(x)/\partial x$ describes the phase gradient along the metasurface. In acoustics, similar wavefront manipulation has been demonstrated using structured phase arrays[17–22]. However, recently some studies[23,24] have shown that this kind of phase gradient metasurface is inherently limited in conversion efficiency for wavefront manipulation, due to impedance mismatch at boundaries. Even for an ideal phase gradient metasurface with infinite resolutions (i.e., $m \to \infty$, where $m$ is the number of unit cells in a superlattice; see below), such a limitation is still present.

A few solutions[23–25] were proposed to successively overcome this inherent limitation to achieve the scattering-free manipulation of anomalous reflected and refracted waves, but the designed metasurfaces require active elements or strong nonlocality, posing challenges for practical implementations[26,27]. To realize extremely anomalous transmission/reflection with perfect efficiency in a passive and lossless structure, bianisotropic metasurfaces[28–30] were proposed and experimentally demonstrated in both electromagnetic and acoustic waves. Alternatively, metagratings[31], periodic structures with a supercell comprising of several subscatters, were suggested to deliver the output wavefront into the desired direction with unity efficiency. However, this method solely works for a specific incidence angle, as the design of the metastructure is well defined for a specific angle. By electrostatically biasing graphene sheets, reconfigurable metagratings[32] can extend the incidence to several discrete angles, but the structures are complex and the working angle is still limited. Therefore, how to realize high-efficient anomalous reflection or/ and anomalous refraction, that can cover a wide incidence in a passive structure, is still an open question.

Essentially, phase gradient metasurfaces are periodic structures with a supercell spatially repeated along the interface, because of folded phase profile[33]. In this way, the GSL is insufficient to determine completely the directions of anomalous reflected or/ and refracted waves, in particular for incident angle beyond the so-called critical angle predicted by the GSL. Instead, it is replaced by another formula involving superlattices[16,19]

$$k_x^{in} = k_x^{r(t)} - nG, \qquad (2)$$

where $G = 2\pi/p$ is the reciprocal lattice vector, and $p$ is period. Both $\xi$ and $G$ commonly share the identical magnitude, yet with different physical origin; the former is introduced by the phase gradient, whereas the latter is caused by the periodicity of grating. Eq. (2) can not only steer a wavefront as expected from the GSL, but can also exhibit other unique features. In fact, in a large number of aforementioned phase gradient metasurfaces, particularly in acoustic metasurfaces[17–22,34–38], anomalous reflection or refraction with high-efficiency were obtained through higher order diffraction. For convenience, in this work we call all

periodic structures with phase gradient as phase gradient metagratings (PGMs). Normally, there are several diffraction channels simultaneously open for a particular incidence and these propagation channels are available for incident wave to depart from PGM. The diffraction mechanism therein is complex and ambiguous, especially in more complicated refractive-type PGMs, since the refractive and reflective diffraction channels are concurrently included. Eq. (2) fails to predict the primary diffraction order of the scattering waves. For instance, for incident angle beyond the critical angle (the $n = 1$ order in Eq. (2)), multiple diffraction channels coexist, and only negative refraction stemming from the $n = -3$ order in Eq. (2) was observed in experiments[19]. The underlying mechanism is still a puzzle.

In this article, we will investigate theoretically and experimentally a passive and lossless refractive-type PGM, and we will show that the designed PGM can enable anomalous reflection and refraction with near unity conversion efficiencies over a wide angle of incidence. Recently, based on loss-induced suppression of higher order diffraction, acoustic asymmetric transmission[39] was demonstrated in the lossy PGMs. Transient simulations revealed that multiple reflections (MRs) are responsible for the energy-loss of higher order diffraction[39], which offered a new insight to explore the uncharted diffraction rule. Starting from the MR effect[39,40], we will reveal the diffraction mechanism of PGMs. It is found that the diffraction order is relevant to the propagation number of MRs (i.e., the number of times the wave travels inside the PGM) and the number of unit cells $m$ of the PGMs. In particular, the transmission and reflection amplitudes of a particular diffraction order are determined by the integer parity of the propagation number. Consequently, the control of transmission and reflection of the diffraction order can be realized by controlling the integer parity, i.e., oddness or evenness (and hereafter referred to simply as parity), of the number of unit cells in the PGMs. Further explorations show that such parity-dependent phenomena are very robust for any $m$, implying that the diffraction law in Eq. (2) should be carefully refined according to the integer parity of $m$. Based on the demonstrated diffraction mechanism, we derive here a new set of formulas that can well explain the complicated diffraction phenomena of our studied PGMs, and can fully predict the parity-dependent perfect anomalous reflection and refraction. The puzzling diffraction phenomena in previous work can also be well understood from our diffraction rule. The experimentally measured results of acoustic waves verify our findings.

## Results

**Models and theory**. To demonstrate our idea, let us start from the metagrating structure shown in Fig. 1a, where the PGM is composed of periodically repeated supercells with lengths of $p$. It should be noted that although this study focuses on acoustic waves, the achieved results are also applicable to the electromagnetic analogs[16]. The whole system is immersed in a background medium of air with density of $\rho_0 = 1.21$ kg m$^{-3}$ and speed of sound $c_0 = 343$ ms$^{-1}$. Figure 1b shows the details of the supercell, which includes $m$ unit cells with widths of $a$ ($=p/m$), and each unit cell is made of sound-hard material (gray area) perforated by a slit (blue area) with a width of $w$. The thickness of the metagrating is $h$. To steer the outgoing wave, the transmitted phase across a supercell covers a complete range of $2\pi$, with a phase gradient of $\xi$. To begin with, we consider effective medium filled in the subwavelength slits. The effective medium is characterized by different effective refractive indices, and the index profile in the $j$th unit cell is given as $n_j = 1 + (j-1)\lambda_0/mh$. For obtaining a specific phase gradient, the period length is set to be constant, and the width of unit cell is determined by the number

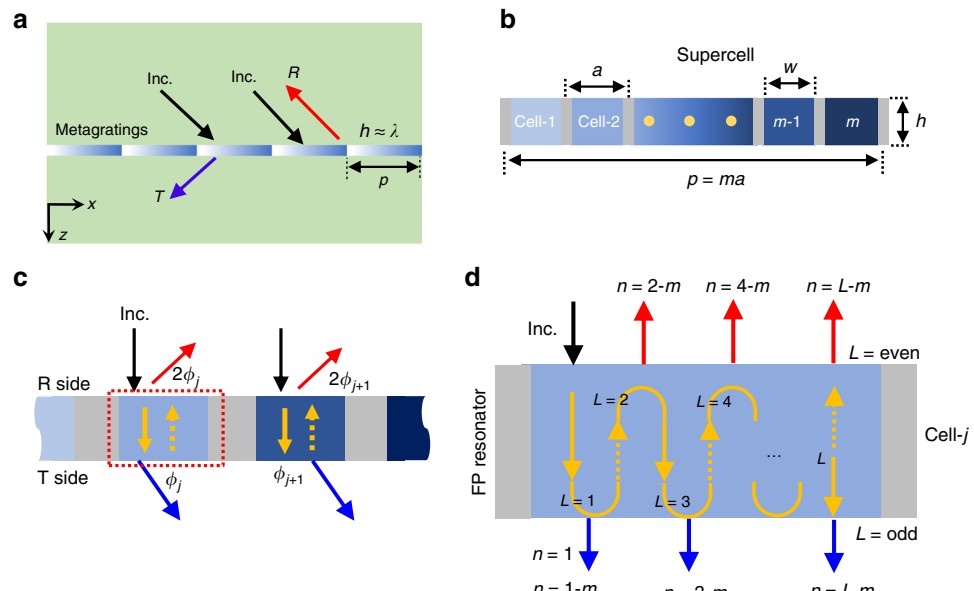

**Fig. 1** Concept of studied metagratings. **a** Schematic diagram of the proposed PGM consisting of periodically repeated supercells. **b** Geometric topology of the supercell composed of $m$ groups of unit cells. The regions in gray are sound-hard materials and the regions with blue colors are gradient index materials for generating gradient phase shift along $+x$-direction. **c** Trajectories of rays propagating in two adjacent unit cells. **d** Sketch map of diffraction mechanism and multiple reflections effect in the $j$th unit cell in (**c**). Each unit cell can be regarded as a Fabry–Perot (FP) resonator, inside which the wave oscillates back and forth $L$ times before reaching a resonance condition that determines the reflection or transmission. The higher order diffraction depends on the propagation number $L$ and the number $m$ of unit cells in a supercell

of unit cells in a supercell. We consider incident wave from air with $k_x^{in} = k_0 \sin \theta_{in}$, where $k_0 = 2\pi/\lambda_0$ is wave vector in air and $\theta_{in}$ is the incident angle. The reflected and transmitted waves obey the diffraction law of Eq. (2), with the maximum diffraction order ($N$, a negative integer) given as, $N =$ roundup $[-2k_0/G] + 1$. Regardless of the direction of the incident wave, i.e., $k_x^{in} \in [-k_0, k_0]$, the existing diffraction orders of the reflected and transmitted waves belong to $n \in [N, 1]$.

**Diffraction mechanism of PGM.** We first provide an intuitive physical picture to reveal the diffraction mechanism. Owing to the sound-hard materials of a PGM (gray area in Fig. 1b), these unit cells could be regarded as acoustic waveguides. The sound-hard material is thick enough to avoid wave coupling between adjacent unit cells. When the width of unit cell is much smaller than the working wavelength (i.e., $a \ll \lambda_0$), only fundamental mode can be supported inside these unit cells. The forward and backward waves propagating in the unit cells interfere to form standing waves stemming from the MR effect of incident rays in the PGM (see the yellow arrows in Fig. 1c). For simplicity, we define the number of times the waves pass through the medium as $L$. When incident rays pass directly through the PGM, i.e., $L = 1$ (see the solid yellow arrows), the phase shift in the $j$th unit cell is $\phi_j = k_0 n_j h$ and the phase difference of adjacent unit cells per period is $\Delta\phi = \phi_{j+1} - \phi_j = 2\pi/m$. By analyzing Eq. (2), the phase gradient of the $n$th diffraction order could be equivalent to $\xi = nG$, accordingly, the phase difference of two adjacent unit cells is expressed as $\Delta\varphi_n = a\xi = 2\pi n/m$. For one-pass propagation, the lowest order $n = 1$ is satisfied for $\Delta\phi = \Delta\varphi_1$, therefore Eq. (2) can be expressed as

$$k_x^{in} = k_x^t - G = k_x^t - \xi, \qquad (3)$$

which is well-known as GSL. In such a case, the incident wave with $k_x^{in} \in [-k_0, k_0 - \xi]$ will follow GSL (the $n = 1$ order), with $k_x = k_0 - \xi$ being the critical momentum. When the incident angle is beyond the critical angle ($k_x^{in} \in [k_0 - \xi, k_0]$), the channel

of the $n = 1$ order will close, and normally the incident wave cannot pass through the PGM via direct transmission. Thereupon, waves will undergo another propagation process ($L = 2$) via internal reflection at the transmitted interface (see the dashed yellow arrows), leading to a phase shift of $2\phi_j$ and a phase difference of $\Delta\phi = 2 \times (2\pi/m)$ at the reflected interface. As the remaining diffraction orders are $n \in [N, 0]$, so $\Delta\varphi_n = 2\pi nm/m \leq 0$. Therefore, it seems that waves cannot couple to these diffraction orders by means of $\Delta\phi = \Delta\varphi_n$. However, when a phase wrap of $2\pi$ is applied to $\Delta\phi$, i.e., $\Delta\phi - 2\pi$ ($2\pi$ phase wrap is enough for wave to couple to higher-order ($N$) in a PGM with unit cells supporting fundamental waveguide modes), the phase difference becomes equivalent. Therefore, when $\Delta\phi - 2\pi = \Delta\varphi_n$, the reflected wave with the $n$-diffraction order will occur (see the red arrows in Fig. 1c); if not, the third time propagation process ($L = 3$) will emerge in unit cells via internal reflection at the reflected interface (see Fig. 1d). When rays reach the transmitted interface, the phase difference is $\Delta\phi = 3 \times (2\pi/m)$. Similarly, if it can meet $\Delta\phi - 2\pi = \Delta\varphi_n$, there will be a transmitted wave of the $n$-diffraction order, otherwise the fourth time propagation ($L = 4$) will happen and so forth (See Fig. 1d).

Generally, if we consider the oscillating wave inside a PGM coupling to the $n$-diffraction order via $L$-time propagation process in unit cells, the corresponding relation can be expressed as $2\pi L/m - 2\pi = 2\pi n/m$, i.e.

$$L = m + n. \qquad (4)$$

As $L > 0$ and the maximum diffraction order is $N$, the number of unit cells is required to meet $m \geq 1 - N$. When $L$ is odd, the incident wave will couple to the corresponding transmitted wave of the higher orders; whereas when $L$ is even, it will couple to the corresponding reflected wave of the higher orders. Consequently, by combining Eqs. (2) and (4),

the diffraction law in a PGM is summarized as

$$\begin{cases} k_x = k_x^t - nG, & (L \text{ is odd}) \\ k_x = k_x^r - nG, & (L \text{ is even}) \end{cases}. \qquad (5)$$

Using Eqs. (3)–(5), the diffraction phenomena in a PGM can be predicted. For the incident wave below the critical angle, the propagation number is $L = 1$, the incident wave will couple to the transmitted wave of the lowest order $n = 1$, which is independent of $m$. For the incident wave beyond the critical angle, MRs happen inside the PGM in turn (i.e., $L = 1 \rightarrow 2 \rightarrow 3...$) and resonance transmission or reflection can be induced when the path length due to MRs reaches the Fabry–Perot condition. If the wave travels through the slab an odd (even) propagation number, strong transmission (reflection) can be generated, with the diffraction order determined by Eq. (5) (see Fig. 1d). Although several diffraction orders are open for the incident wave, the maximum diffraction order is preferential owing to minimum propagation number that corresponds to minimum geometric path length. Furthermore, if one designs a PGM with odd and even unit cells, caused by the parity transition of the propagation number, the transmission and reflection of the diffraction order can be reversed.

**Analytical and numerical demonstration.** Although the above revealed diffraction mechanism and associated diffraction rule are very simple, they are indeed powerful for making complex diffraction phenomena clear. Without loss of generality, we take PGMs with $\xi = k_0$ to verify this point, in which the maximum diffraction order is $N = -1$ and the critical angle of is $\theta_1 = 0°$. When $\theta_{in} < \theta_1$, the propagation number is $L = 1$, it is mainly the transmitted wave following GSL (the $n = 1$ order). While for $\theta_{in} > \theta_1$, there are two diffraction orders, i.e., the $n = 0$ order and the $n = -1$ order. As we have discussed in Fig. 1d, the higher

diffraction order is preferential owing to the minimum propagation number. Hence, for $\theta_{in} > \theta_1$, the effective diffraction order is the $n = -1$ order and the corresponding propagation number of PGM with $m$ unit cells is $L = m - 1$. Based on Eqs. (4) and (5), when $m$ is odd, e.g., $m = 3$, the propagation number $L$ is even, which leads to the reflection of the $n = -1$ order (see the equi-frequency contour in Fig. 2a). On the other hand, when $m$ is even, e.g., $m = 4$, the propagation number $L$ is odd, which results in the transmission of the $n = -1$ order (see the equi-frequency contour in Fig. 2b). To demonstrate above theoretical prediction, numerical simulations are performed using COMSOL MULTI-PHYSICS. The simulated field patterns of the PGMs with three and four unit cells are respectively displayed in Fig. 2c, d, where these two metagratings share identical phase gradient, since the period length is constant, i.e., $p = \lambda_0$. When $\theta_{in} = -30°$, which is below the critical angle, the incident waves in both cases pass through the PGMs following the $n = 1$ order (see the lower plots of Fig. 2c, d). However, when $\theta_{in} = 30°$, beyond the critical angle, the incident wave is reflected back for the PGM with $m = 3$ (see the upper plot in Fig. 2c), and passes through the PGM with $m = 4$ (see the upper plot in Fig. 2d). In both cases, the scattered waves follow the $n = -1$ diffraction order with nearly perfect conversion efficiency (see the arrows in Fig. 2). Therefore, the theoretical prediction based on Eqs. (4) and (5) is well demonstrated from numerically simulated acoustic field patterns. In addition, by observing the equicontour in Fig. 2a, when the incident angle is within the critical angle, the anomalous transmission can occur by following $k_x^t = k_x + \xi$. While for the incident angle beyond the critical angle, the PGM will generate an equivalent tangential momentum of $-\xi$, the anomalous reflection will take place by obeying $k_x^r = k_x - \xi$. Hence, bounded by the critical angle, the anomalous reflection and transmission can simultaneously exist in a single PGM, which enables potential design for multifunctional acoustic planar devices.

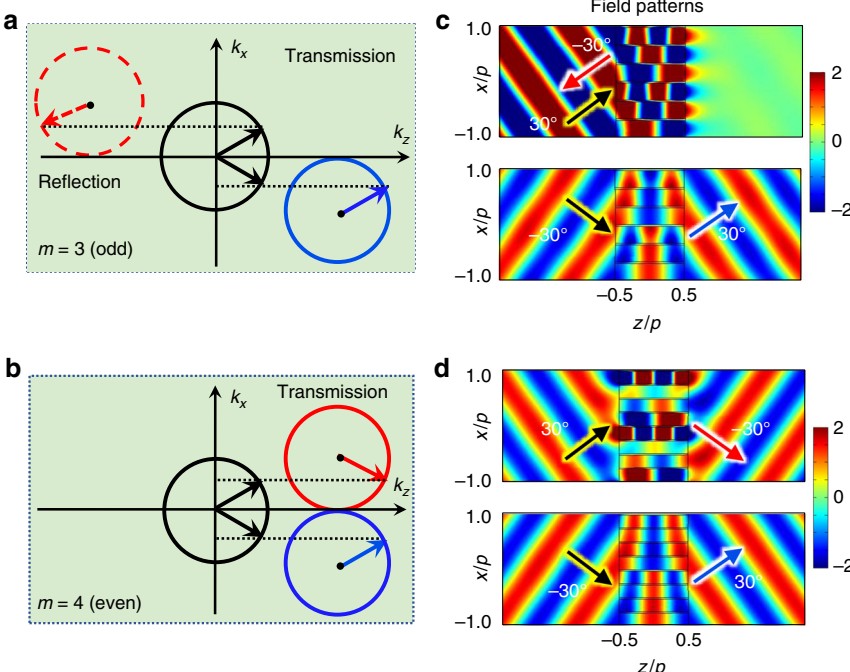

**Fig. 2** Parity-dependent phenomena. **a, b** The equifrequency contours of the PGMs ($\xi = k_0$) with odd unit cells and even unit cells, respectively, where the black circles are the equifrequency contours of incident wave in air, the blue circles are the transmission contours of the $n = 1$ order and the solid (dashed) red circle is the transmission (reflection) contour of the $n = -1$ order. **c, d** The simulated acoustic total field patterns of the PGMs ($\xi = k_0$) with three cells and four unit cells, respectively. In each case, the upper (lower) plot is the case of $\theta_{in} = 30°$ ($\theta_{in} = -30°$). All the arrows represent propagation directions. In simulations, $p = h = \lambda_0$, $a = 0.9w$ and $n_j = \rho_j$

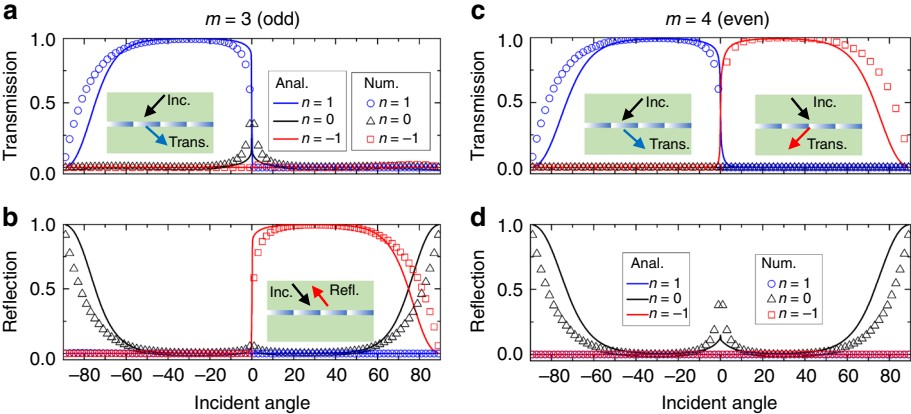

**Fig. 3** The relationship between transmission/reflection and incident angle of all diffraction orders in the PGMs with $\xi = k_0$. **a, b** The PGM with $m = 3$. **c, d** The PGM with $m = 4$. The solid lines are analytical results, and the symbols represent numerical results

In fact, the reflection (transmission) of the $n = -1$ order for the PGM with $m = 3$ ($m = 4$) not only happens at $\theta_{in} = 30°$, but occurs in a wider incident range, which can be observed from the equi-contours in Fig. 2. To quantify angular performance of the PGMs, we analytically and numerically show the relationship between the transmission/reflection of the diffraction orders and the incident angle in Fig. 3, where the analytical results are described by the curves and the numerical results are indicated by the symbols. The analytical results are obtained based on the coupled mode theory[18,39] (details shown in Supplementary Note 1), which agree well with numerical results except for very steep incident angles. For the case of the PGM with $m = 3$ (see Fig. 3a, b), more than 90% transmission of the $n = 1$ order is observed for $\theta_{in} \in [-60°, 0°]$ and the reflection of the $n = -1$ order is higher than 90% for $\theta_{in} \in [0°, 60°]$. In addition, for the normal incidence, that is, at the critical angle, there is an odd propagation number of $L = 3$ for the $n = 0$ order, bringing about higher transmission of the $n = 0$ order (see the black data in Fig. 3a). For the case of the PGM with $m = 4$ (see Fig. 3c, d), similar behavior is observed for $\theta_{in} \in [-60°, 0°]$. For angles above the critical angle ($\theta_{in} \in [0°, 60°]$), however, the reflection mode is reversed to transmission mode due to integer parity of the cell number. Furthermore, it is an even propagation number of $L = 4$ for the $n = 0$ order, therefore there is higher reflection of the $n = 0$ order at the critical angle (see the black data in Fig. 3d), which is opposite with that in Fig. 3a. For the incident angle near ±90°, owing to intrinsic limitation of PGMs[29,30], the coupling efficiency between incident wave and the $n = 1/n = -1$ order is extremely low and stronger specular reflection appears (see black data in Fig. 3b, d). In addition, to further demonstrate Eqs. (4) and (5), as a more complicated case, PGMs with $\xi = 0.6k_0$ are used to reveal similar reversal phenomena of the diffraction orders, shown in Supplementary Figs. 1 and 2, and Supplementary Note 2.

**Design of PGMs and experimental verifications**. To further confirm the diffraction behavior of the PGMs ($\xi = k_0$) with odd/even number of unit cells, we utilize zigzag microstructures to design two groups of PGMs at 4.0 kHz: one is a PGM with three unit cells and the other has four unit cells. In each case, the transmissions of these designed unit cells are nearly unity and the phase differences between two adjacent cells are $\Delta\phi = \phi_{j+1} - \phi_j = 2\pi/m$ ($m = 3$ and 4). The Supplementary Figs. 3 and 5, and Supplementary Notes 3 and 4 show the physical dimensions of the final designs and design details. The fabricated samples of the two PGMs are shown in Fig. 4a, where one period of the PGM with $m = 3$ ($m = 4$) is highlighted by the red (blue) box (see the inset). The experimental setup is shown in Fig. 4b. For the designed PGM with

$m = 3$, we numerically show the corresponding relationship between the transmission/reflection of the main diffraction orders ($T_1$, $T_0$ and $R_{-1}$) and the incident angle in Fig. 4c, where the transmission/reflection agrees well with that in the ideal case of Fig. 3. In the experiments, to measure the angular performance of the designed PGMs, the Gaussian beam from the speaker array is incident from $\theta_{in} = -60°$ to $\theta_{in} = 60°$ with a step of 15°, and the measured results denoted by the stars are also displayed in Fig. 4c. While the measured result has some deviation in amplitude from the numerical result, the variation tendency of the curves agrees well with each other. Indeed, anomalous reflection and transmission can simultaneously exist in such a single PGM. To clearly show the reflection performance of the PGM with $m = 3$, the simulated scattered field, including reflected field and transmitted field of incident beam with $\theta_{in} = 30°$ (for other angles, see Supplementary Fig. 4) is shown in Fig. 4d, where a strong reflected wave towards the opposite direction with the incident wave is seen and the transmitted wave is much weaker. The experimentally measured scattered field shows the identical result (see Fig. 4e).

For the designed PGM with $m = 4$, the corresponding relationship between the transmission/reflection of the main diffraction orders ($T_1$, $T_{-1}$, and $R_0$) and the incident angle is shown in Fig. 4f, with transmission/reflection agreeing well with prediction (Fig. 3). The corresponding measured result is displayed in Fig. 4f, where there are some small discrepancies between the numerical and measured results, but the overall trend in both cases is consistent. In addition, the field patterns of simulated and measured scattered waves for $\theta_{in} = 30°$ (for other angles, see Supplementary Fig. 6) are respectively shown in Fig. 4g, h, where both results reveal that strong transmitted waves appear and reflected waves are considerably weaker. Therefore, the parity design of the PGMs can effectively manipulate the switching of reflection and transmission of the higher order diffraction, enabling more flexibility in the design of acoustic planar devices.

**Robust feature of parity-dependent transmission and reflection**. We would also like to point out that the phenomenon of anomalous reflection and refraction in the PGMs with parity design is very robust, depending only on the parity of the number $m$ of unit cells. The reversal phenomenon could be observed in a PGM with parity design, even with large $m$, as long as the wave coupling between adjacent unit cells is negligible. To demonstrate this robust feature, we consider a specific case of a PGM with $\xi > k_0$ as an example, and analyze the behavior at normal incidence. In this way, only the transmission and reflection of the $n = 0$ order[41] need to be taken into consideration. After some mathematical derivations (see Supplementary Note 5), the

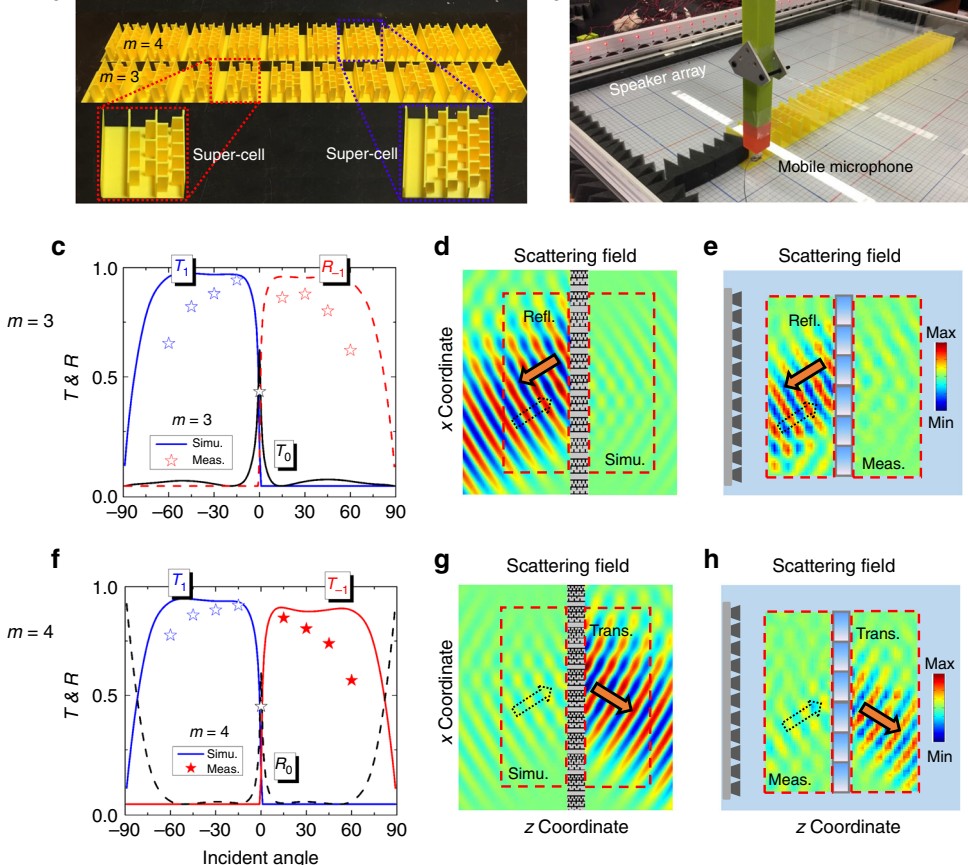

**Fig. 4** Experimental setup and results. **a** Photograph of the fabricated samples. **b** Photograph of experimental setup. **c** The relationship between transmission/reflection ($T_1$, $T_0$ and $R_{-1}$) and the incident angle for the designed PGM with $m = 3$. **d**, **e** The simulated and measured scattered pressure field patterns for incident beam with $\theta_{in} = 30°$ bumping on the designed PGM with $m = 3$, respectively. **f** The relationship between transmission/reflection ($T_1$, $T_{-1}$, and $R_0$) and the incident angle for the designed PGM with $m = 4$. **g**, **h** The simulated and measured scattering pressure field patterns for incident beam with $\theta_{in} = 30°$ bumping on the designed PGM with $m = 4$

corresponding transmission and reflection coefficients for the $n = 0$ order are respectively given as

$$r_0 = \frac{|\zeta_1|^2 - |\zeta_2|^2}{\zeta_1^2 - \zeta_2^2}, \quad t_0 = \frac{\zeta_1\zeta_2^* - \zeta_1^*\zeta_2}{\zeta_1^2 - \zeta_2^2}, \quad (6)$$

where $\zeta_1 = (\tilde{g} - 1)\sum_{j=1}^{m} 1/(u_j^2 - 1)$, $\zeta_2 = (1 - \tilde{g})\sum_{j=1}^{m} u_j/(u_j^2 - 1)$, $\tilde{g}_1 = 2g_1^2/(g_1^2 - \gamma_1)$, $g_1 = \mathrm{sinc}(Gw/2)$ and $u_j = \exp(i\phi_j)$ is phase shift in the $j$th unit cell in $u$-complex plane. From Eq. (6), we know that the reflection and transmission are only determined by two factors: (i) the coefficient $\tilde{g}_1$, which is related to the geometry structure of a PGM and is a constant for a fixed configuration. (ii) the sums of $Y_1 = \sum_{j=1}^{m} 1/(u_j^2 - 1)$ and $Y_2 = \sum_{j=1}^{m} u_j/(u_j^2 - 1)$, which are highly dependent on the phase distribution $\phi_j$ in $u$-complex plane. When $m$ is odd, the phase distribution of $\phi_j$ is asymmetric (see Fig. 5a), which results in $|Y_1| = |Y_2|$. When $m$ is even, the phase distribution of $\phi_j$ is symmetric (see Fig. 5b), and $Y_2 = 0$. The detailed mathematical derivation is shown in Supplementary Note 5. With these results, Eq. (6) becomes

$$r_0 = 0, \quad t_0 = \exp(-i\varphi_T), \quad m \text{ is odd}; \quad (7)$$

$$r_0 = \exp(-i\varphi_R), \quad t_0 = 0, \quad m \text{ is even}; \quad (8)$$

where $\varphi_T = \arg(\zeta_1) + \arg(\zeta_2)$ and $\varphi_R = 2\arg(\zeta_1)$, giving rise to a perfect transmission for odd $m$ and a perfect reflection for even $m$.

The results are consistent with these from the theoretical prediction summarized in Eqs. (4) and (5). Based on the generalized theoretical formulas in Eqs. (S11)–(S14), Fig. 5c, d displays the results of $R = \sum_n |r_n(m)|$ and $T = \sum_n |t_n(m)|$ with $n = 0$, respectively, which agree well with the approximate results of Eqs. (7) and (8). Therefore, it is analytically confirmed that even for larger $m$, the reversal phenomenon of almost perfect transmission and reflection is preserved, implying the parity-dependent feature is quite robust.

## Discussion

In conclusion, through a combination of analytical calculations and numerical simulations, we have revealed the governing diffraction mechanism of PGMs from the perspective of MRs. We find that the integer parity of the PGMs plays a pivotal role in the higher order diffraction for incident waves beyond the critical angle. To be more precise, the parity transition in the designed unit cells of a PGM enables the relevant odd/even transition of propagation number in the unit cells, which induces the reversal of transmission and reflection for a particular diffraction order. To demonstrate our findings, two acoustic PGMs ($\xi = k_0$) with three and four unit cells are designed using zigzag microstructures, and the reversal phenomenon of higher order diffraction is clearly demonstrated in experiments. In particular, the coexistence of anomalous reflection and anomalous transmission, depending on a critical angle, is achieved in a single PGM with an odd number of unit cells. Compared with previous works in both acoustic waves and electromagnetic waves, the diffraction

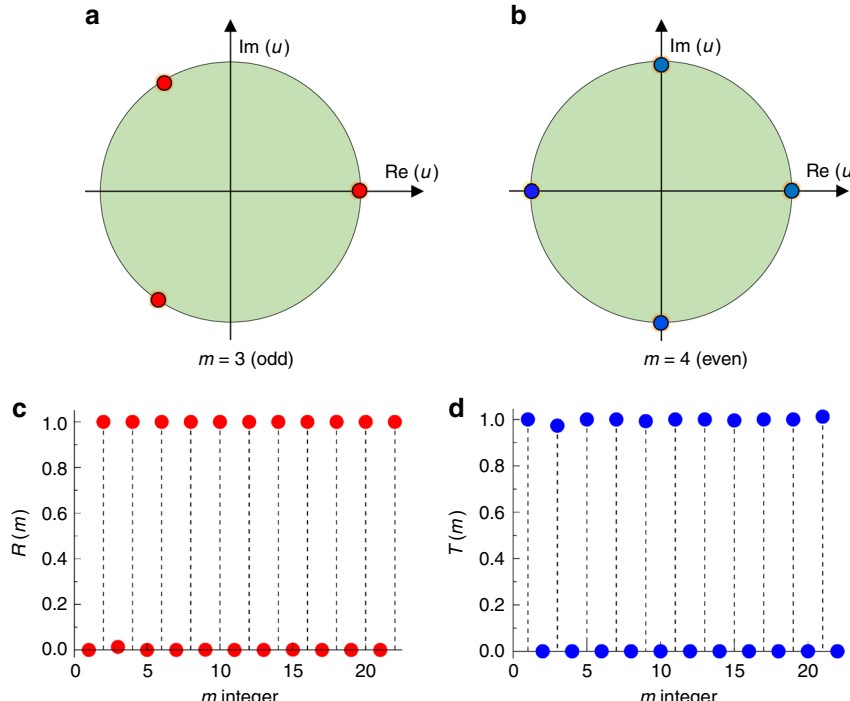

**Fig. 5** Parity-dependent transmission and reflection and robust performance. **a**, **b** The phase distributions in the *u*-complex plane for $m = 3$ (odd) and $m = 4$ (even), respectively. **c**, **d** The reflection (**c**) and transmission (**d**) vs. integer $m$ in a PGM with $\xi = 1.5k_0$ based on coupled mode theory

mechanism proposed here can comprehensively explain almost all the known diffraction behaviors in the metagratings with phase gradient. While our system is designed to work at a specific frequency, the parity-dependent behavior can be observed in a certain bandwidth as the phase gradient along the metagrating is preserved, leading to some tolerances in the frequency response (see Supplementary Fig. 7 and Supplementary Note 6). In addition, if a larger $m$-integer is designed for the lossy metagratings, the higher diffraction orders will undergo more round-trips, along with more absorption[40]. As a result, the parity-dependent scattering behavior of the higher diffraction orders will gradually disappear as "$m$" increases. We believe that our proposed diffraction mechanism can become a new paradigm for the design of acoustic/electromagnetic PGMs and open up new wave manipulation capabilities based on the versatile platform that can offer. For instance, due to achieved anomalous refraction and reflection, our work enables more systematic design of functional planar devices, such as asymmetric and wide-angle absorbers[39,40], multifunctional metagratings[42], omnidirectional reflector[43–45]. Alternatively, inspired by the phenomenon that an incident wave can be totally transmitted or reflected by a disordered slab[46–48], one can design a metagrating of disorder with a properly designed combinations of integer m, which might enable some new effects associated with disorder-induced transition.

## Methods

**Numerical simulations**. The full wave simulations are performed using COMSOL Multiphysics Pressure Acoustics module. In Fig. 2, the plane wave is incident on the PGM consisting of two supercells, the upper and lower walls are set as periodic boundary conditions and perfectly matched layers (PMLs) are used in the left and right sides to reduce the reflection. In Fig. 4, a spatially modulated Gaussian wave is incident on the designed PGM with 20 supercells, and the surrounding regions are PMLs. The normalized transmission and reflection efficiencies of the diffraction orders are numerically obtained from the port analysis of COMSOL RF module, where the acoustic profiles are replaced by their optical analogs.

**Experimental apparatus**. The samples were fabricated with fused deposition modeling in three dimensional printing and the printed material is acrylonitrile

butadiene styrene plastic with density of 1180 kg m$^{-3}$ and speed of sound 2700 ms$^{-1}$. As the characteristic impedance of the plastic is much larger than that of air, the walls can be considered as acoustically rigid. The fabricated PGM consists of ten supercells and is placed in a two-dimensional waveguide for the measurement. A loudspeaker array with 28 speakers emits a Gaussian modulated beam to the PGM and the reflected and transmitted field is scanned using a moving microphone with a step of 2.0 cm. The acoustic field at each spot is then calculated using Fourier Transform. The overall scanned area is 90 cm by 30 cm and the signal at each position is averaged out of four measurements to reduce noise. The transmission/ reflection efficient is calculated by performing Fourier Transform along a line right behind/in front of the PGM.

## Data availability

The data that support the findings of this study are available from the corresponding author upon reasonable request.

## Code availability

The code used for the analyses will be made available upon e-mail request to the corresponding author.

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

## Acknowledgements

This work was supported by the National Natural Science Foundation of China (grant Nos. 11604229, 11774252, and 11874311), the Natural Science Foundation of Jiangsu Province (grant Nos. BK20161210 and BK20171206), a Multidisciplinary University Research Initiative grant from the Office of Naval Research (N00014-13-1-0631), an Emerging Frontiers in Research and Innovation grant from the National Science Foundation (grant No. 1641084), the Project funded by China Postdoctoral Science Foundation (grant No. 2018T110540), Hong Kong Research Grants Council (AoE/P-02/12), and the Fundamental Research Funds for the Central Universities (grant No. 20720170015). Fu would like to thank the start-up fund support from Nanjing University of Aeronautics and Astronautics, Xu would like to thank the support from the Collaborative Innovation Center of Suzhou Nano Science and Technology at Soochow University, and Gao thanks the support from the Qing Lan project, "333" project (BRA2015353) and PAPD of Jiangsu Higher Education Institutions. We also thank the helpful discussions with Prof. Z.-Q. Zhang from Hong Kong University of Science and Technology.

## Author contributions

Y.X. and Y.F. conceived the idea. Y.F., C.S., Y.C., and Y.X. performed the theoretical calculation and numerical simulations. C.S. and S.A.C. fabricated the samples and performed experiments. L.G. and H.C. helped with the theoretical interpretation. Y.X., C.T.C., and S.A.C. supervised the project. All authors discussed the results and prepared the paper.
