## [Peer Review File · Nature Communications]

Reviewers' Comments:

Reviewer #1:

Remarks to the Author:

In this article, the authors investigate theoretically, numerically and experimentally the transmission/reflection of acoustic waves through a metagrating. More specifically, they demonstrate how they can switch from a fully reflecting metasurface to a fully transmitted one just by playing with the number of sub-elements constituting their super-cell. Their work focuses on the case of acoustic waves but it can easily be generalized to any type of waves. When I see a complete work starting from a theoretical description down to an experimental realization through the use of numerical simulations to confirm the claims, my role as a reviewer is only to congratulate the authors and add some inputs to maybe broaden the readership of the paper. As you understand I recommend the publication of the paper in Nature Communications almost as it is, since the results are clear and well described.

I only have few comments that I would like the authors to answer:

1. First, the results presented in the paper assume from the beginning a very specific type of metagrating which contains two interfaces. This allows to count for the round-trips between those two interfaces, and notably the accumulated phase between the two interfaces plays a crucial role. Since the authors have presented a relatively extensive review of the different metagratings, I wonder if everything can generalize to any type of metasurface. Notably, I do not think that it can apply to single layer metasurfaces... Or, can we obtain similar results with metagratings consisting of two almost touching parallel metasurfaces? In the latter case the phase at reflection on the two sheets could play the role of the accumulated phase through complicated paths of the current proposal.

2. My second remark concerns the spectral behavior of such a concept. Indeed, all of the results are presented for a single frequency (indeed the spatial gradient is chosen for this frequency) but I am wondering what happens spectrally. Are the effects completely lost or can we define a spectral quality factor of the device? Is it a very narrowband phenomenon or it does it occur for a broad spectral range as we need most of acoustical applications?

3. Another question concerns the effect of losses. The experimental points clearly reveal that when adding losses (and when you build tortuous paths you have obviously viscous losses) that the transmission and reflection do not reach the desired value of 1. I expect this effect to become more and more dramatic by increasing the m -integer which will make the figure 5 irrelevant. Could the author discuss this aspect in the paper?

4. Eventually, I would love if the authors could make some connections with the transport theories of electrons (Dorokhov, Solid State Commun. 51, 381 (1984); Y. Imry, Europhys. Lett. 1, 249 (1986); or recently brought to the wave community Gerardin, PRL 113(17), 173901 (2014)). Those concepts concern the idea of the opening and the closing of transmission channels through disordered slab. In their case they have many channels and they show that ultimately they can classify the channels as fully transmitting ones, and fully reflecting ones. In the present case, the number of channels is reduced and it becomes possible to count all of them.

Reviewer #2:

Remarks to the Author:

The authors investigated analytically and experimentally a refractive-type metagrating and show that anomalous reflection and refraction with almost unity conversion efficiency can be achieved within a wide range of incident angles. The reviewer thinks that the work is interesting and well motivated. The study is well conducted, results are well presented and the manuscript is overall

very well written. The author should address one minor concern on page 4 section "Diffraction mechanism of PGM" line 143: there is an unclear symbol λ_0 . The reviewer suggests acceptance for publication after this minor correction.

Response to the reviewer's comments

We thank the reviewers for their comments and suggestions which are helpful to further improve our manuscript. In the resubmitted manuscript, we have made revisions accordingly. Below is our response to the reviewer's comments.

Reviewers' comments:

Reviewer #1 (Remarks to the Author):

In this article, the authors investigate theoretically, numerically and experimentally the transmission/reflection of acoustic waves through a metagrating. More specifically, they demonstrate how they can switch from a fully reflecting metasurface to a fully transmitted one just by playing with the number of sub-elements constituting their super-cell. Their work focuses on the case of acoustic waves but it can easily be generalized to any type of waves. When I see a complete work starting from a theoretical description down to an experimental realization through the use of numerical simulations to confirm the claims, my role as a reviewer is only to congratulate the authors and add some inputs to maybe broaden the readership of the paper. As you understand I recommend the publication of the paper in Nature Communications almost as it is, since the results are clear and well described.

Reply: Thanks for the reviewer's positive comments.

I only have few comments that I would like the authors to answer:

1. First, the results presented in the paper assume from the beginning a very specific type of metagrating which contains two interfaces. This allows to count for the round-trips between those two interfaces, and notably the accumulated phase between the two interfaces plays a crucial role. Since the authors have presented a relatively extensive review of the different metagratings, I wonder if everything can generalize to any type of metasurface. Notably, I do not think that it can apply to single layer metasurfaces... Or, can we obtain similar results with metagratings consisting of two almost touching parallel metasurfaces? In the latter case the phase at reflection on the two sheets could play the role of the accumulated phase through complicated paths of the current proposal.

Reply: The reviewer has raised a good question. There are several types of metasurfaces or metagratings based on different design approaches or mechanisms. Different from other single-layer metasurfaces, our metasurface (metagrating) systems are featured with phase gradient unit cells bounded by impenetrable walls, which enables that the guided mode in each unit cell can independently oscillate back and forth so that the related phase accumulation can be directly obtained through the geometric paths. If such a design mechanism is extended to other types of metasurfaces (e.g., metallic gratings with phase gradient), similar results could be realized.

With regard to the case of two parallel metagratings with a small gap, it turns into a more complicated case that is beyond the scope of this study. We do not think similar scattering processes could be easily produced through guided modes oscillating inside unit cells, as surface wave coupling stemming from higher diffraction orders will play an important role in the opened gap. Therefore, to be honest, we are not sure whether the reported parity-dependent phenomenon could be easily reproduced in this case, but it is worth further exploring. In the future, we will perform some research on this problem and also expect to reveal the unknown physics therein.

2. My second remark concerns the spectral behavior of such a concept. Indeed, all of the results are presented for a single frequency (indeed the spatial gradient is chosen for this frequency) but I am wondering what happens spectrally. Are the effects completely lost or can we define a spectral quality factor of the device? Is it a very narrowband phenomenon or it does it occur for a broad spectral range as we need most of acoustical applications?

Reply: As the phase gradient unit cells are designed at the targeted frequency, a well-designed metagrating only works in a narrowband. If the working frequency deviates from the targeted one, the required phase and transmission profiles in these individual unit cells are also compromised. As a result, the transmission/reflection efficiencies of the desired diffraction orders will reduce owing to the generation of other undesired diffraction orders. Therefore, the major effects are not completely lost for a slight frequency shift.

However, it is difficult to quantify the device using a certain quality factor, as the practical functionality of the device is highly dependent on the monolithic response of these designed unit cells. This response usually has intricate phase and transmission response in the spectrum, and could vary for different types of designs. Therefore, the performance of the designed device only can be numerically/experimentally tested case by case, which is quite common in the subject of phase-gradient metasurfaces.

For example, we show the transmission and reflection of the designed metagrating with $m=3$ at other wavelengths. The required phase and transmission profiles are maintained approximately from $0.95\lambda_0$ to $1.1\lambda_0$ (see panels (a) and (b)), the desired performance of the device can be preserved, e.g., see the result in panel (e). For a bigger change in wavelength (e.g., $0.9\lambda_0$ or $1.2\lambda_0$), both the phase and transmission profiles deviate from the desired results, and the performance further reduces (see the panels (d) and (f)). In the smaller wavelength regime (e.g., $0.8\lambda_0$), transmission profiles are good, but phase profiles deviate from the desired results, leading to a compromised performance (see the panel (c)).

Figure-The frequency response (Fig. S7)

To address this problem, we add some discussion in the conclusion part, which reads as,

“While our system is designed to work at a specific frequency, the parity-dependent behavior can be observed in a certain bandwidth as the phase gradient along the metagrating is preserved (see the

Supplementary Fig. S7 and the Supplementary Note 6), leading to some tolerances in the frequency response.”

3. Another question concerns the effect of losses. The experimental points clearly reveal that when adding losses (and when you build tortuous paths you have obviously viscous losses) that the transmission and reflection do not reach the desired value of 1. I expect this effect to become more and more dramatic by increasing the m-integer which will make the figure 5 irrelevant. Could the author discuss this aspect in the paper?

Reply: We agree with the reviewer. When the number of the unit cells is gradually increased, based on the formulas of $L=m+n$, the geometric paths of the higher diffraction order (i.e., $n=-1$ in Fig.5) will increase accordingly. As a longer propagation path can lead to more absorption, the amplitude of higher diffraction order will gradually reduce. Therefore, the parity-dependent scattering behavior of the higher diffraction order will disappear if a larger m-integer is considered in practice. This, however, could lead to other interesting phenomena such as enhanced absorption since the effect of losses is boosted (e.g., Phys. Rev. Appl. 9, 054009 (2018)).

We have added several sentences to discuss this point in the conclusion part, which reads as,

“In addition, if a larger m-integer is designed for the lossy metagratings, the higher diffraction orders will undergo more round-trips, along with more absorption [40]. As a result, the parity-dependent scattering behavior of the higher diffraction orders will gradually disappear as “m” increases.”

4. Eventually, I would love if the authors could make some connections with the transport theories of electrons (Dorokhov, Solid State Commun. 51, 381 (1984); Y. Imry, Europhys. Lett.1, 249 (1986); or recently brought to the wave community Gerardin, PRL 113(17), 173901 (2014)). Those concepts concern the idea of the opening and the closing of transmission channels through disordered slab. In their case they have many channels and they show that ultimately they can classify the channels as fully transmitting ones, and fully reflecting ones. In the present case, the number of channels is reduced and it becomes possible to count all of them.

Reply: Many thanks to the reviewer for his/her constructive suggestions and for communicating the results of these papers. The connection between them is now discussed at the end of the revised version, which reads as

“For instance, inspired by the phenomenon that an incident wave can be totally transmitted or reflected by a disordered slab [46-48], one can design a metagrating of disorder with a properly designed combinations of integer m, which might enable some new effects associated with disorder-induced transition.”

Accordingly the mentioned papers have been quoted properly.

Reviewer #2 (Remarks to the Author):

The authors investigated analytically and experimentally a refractive-type metagrating and show that anomalous reflection and refraction with almost unity conversion efficiency can be achieved within a wide range of incident angles. The reviewer thinks that the work is interesting and well motivated. The study is well conducted, results are well presented and the manuscript is overall very well written. The author should address one minor concern on page 4 section “Diffraction mechanism of PGM” line 143: there is an unclear symbol a λ_0 . The reviewer suggests acceptance for publication after this minor correction.

Reply: We appreciate the reviewer's constructive comments. The correct symbol should read $a \ll \lambda_0$.

This error has been corrected in the revised version.

Reviewers' Comments:

Reviewer #1:

Remarks to the Author:

The authors have properly answered all of my concerns. I totally reommend publication of the paper.

Response to the reviewer's comments

Reviewer #1 (Remarks to the Author):

The authors have properly answered all of my concerns. I totally recommend publication of the paper.

Reply: Great thanks for the reviewer's positive comments.